# More Insights about the Efficacy of Copper Ion Treatment on *Mycobacterium avium* subsp. *paratuberculosis* (MAP): A Clue for the Observed Tolerance

**DOI:** 10.3390/pathogens11020272

**Published:** 2022-02-19

**Authors:** Carlos Tejeda, Pamela Steuer, Marcela Villegas, Angelica Reyes-Jara, Esperanza C. Iranzo, Reydoret Umaña, Miguel Salgado

**Affiliations:** 1Instituto de Medicina Preventiva Veterinaria, Facultad de Ciencias Veterinarias, Universidad Austral de Chile, Valdivia 5090000, Chile; carlos.tejeda@alumnos.uach.cl (C.T.); pamela.steuer@uach.cl (P.S.); marcela.villegas@alumnos.uach.cl (M.V.); reydoretumana@uach.cl (R.U.); 2Escuela de Graduados, Facultad de Ciencias Veterinarias, Universidad Austral de Chile, Valdivia 5090000, Chile; 3Laboratorio de Microbiología y Probióticos, Instituto de Nutrición y Tecnología de los Alimentos, Universidad de Chile, Santiago 8330015, Chile; areyes@inta.uchile.cl; 4Laboratorio de Manejo y Conservación de Vida Silvestre, Instituto de Ciencia Animal y Programa de Investigación Aplicada en Fauna Silvestre, Facultad de Ciencias Veterinarias, Universidad Austral de Chile, Valdivia 5090000, Chile; esperanza.iranzo@gmail.com

**Keywords:** *Mycobacterium avium* subsp. *paratuberculosis*, copper ion device, efficacy, MGIT culture, qPCR, physicochemical properties

## Abstract

Background: Scientific evidence is scarce for the antimicrobial effect of copper on bacteria characterized as more resistant. Using *Mycobacterium avium* subsp. *paratuberculosis* (MAP), a highly resistant microorganism, as a pathogen model, copper ion treatment has shown a significant bactericidal effect; however, the sustainability of MAP against copper toxicity was also reported in several studies. Accordingly, the present study aimed to evaluate the impacts of copper on MAP. Methodology: This study considered physicochemical properties and copper concentration in a buffer since it could modulate MAP response during the application of copper treatment. Results: Despite the efficacy of copper ions in significantly reducing the MAP load in Phosphate Buffered Saline, some MAP cells were able to survive. The copper concentration generated by the copper ion treatment device increased significantly with increasing exposure times. MAP bacterial load decreased significantly when treated with copper ions as the exposure times increased. An increase in pH decreased oxygen consumption, and an increase in conductivity was reported after treatment application. Conclusions: Even with higher concentrations of copper, the efficacy of MAP control was not complete. The concentration of copper must be a key element in achieving control of highly resistant microorganisms.

## 1. Introduction

Copper is a chemical element that, at low concentration, plays an essential role in animal and plant cell metabolism [1]. At present, over 30 types of copper-containing proteins essential for life (e.g., ceruloplasmin, superoxide dismutase, cytochrome oxidase), are known [1]. 

The importance of copper has been understood since its antibacterial properties were discovered in the 5th and 6th millennium BC [2,3]. The scientific findings that confirm the effectiveness of copper on the integrity of Gram-negative and Gram-positive bacteria, viruses, and fungi are very well documented [4,5,6,7,8]. 

Most of the research has focused on the sanitizing properties of copper and bacterial death mediated by direct contact with surfaces of pure copper or its alloys [3]. The antibacterial effect of copper surfaces on bacteria such as *Escherichia coli* and *Clostridium difficile* [5,9] has been tested under different conditions. It has even been demonstrated on potent drug-resistant nosocomial pathogens such as *Enterobacter* spp., *Klebsiella pneumoniae*, *Pseudomonas aeruginosa*, *Acinetobacter baumannii*, and *Mycobacterium tuberculosis* [10,11].

Although different antibacterial copper preparations, such as copper sulfate (CuSO_4_) and copper surfaces [3,12], have been used, the use of copper surfaces has been the strategy that has gained the broadest scientific support. Another less known antibacterial copper strategy is the use of copper particles, specifically nanoparticles [13]. 

In a more novel way, we have reported that copper ions, in a liquid matrix [14,15], are highly effective against *Mycobacterium avium* subsp. *paratuberculosis* (MAP), one of the most fastidious members of the *Mycobacterium* genus that is also highly resistant to both physical and chemical agents [16,17,18,19] and the causative agent of paratuberculosis in ruminants [20] with zoonotic potential [21]. Our experiment used a glass receptacle copper device in which copper plates of high purity (99%) were immersed in a liquid matrix and were stimulated with electricity [14,15]. This method allows a greater release of ions than would be the case without electricity, as shown in a study, in which copper plates stimulated with electricity resulted in reduced growth of *E. coli* bacteria by 79.5 ± 2.34% [22]. Interestingly, some MAP strains managed to tolerate this novel copper treatment [14,15]. 

The concept of tolerance is defined as the ability of a microorganism to survive but neither grow nor die in the presence of a bactericidal antimicrobial agent [23], and in practical terms, a measure of tolerance is the minimum bacteriostatic concentration. We believe that the concentration of copper in buffer, as well as the physicochemical characteristics of this liquid matrix once the treatment with copper ions has been applied, are key elements to understand the efficacy of this treatment. Therefore, in the present study, we sought to deepen our understanding of the efficacy of this copper treatment on MAP cells and of the physicochemical properties of the liquid matrix in which MAP cells had been suspended, since these could modulate MAP inhibition and/or its response during the application of this novel copper treatment. Through this undertaking, we hoped to gain further insights into the efficacy of this novel copper-based treatment on one of the most drug-resistant potential zoonotic pathogens—MAP.

## 2. Results

### 2.1. Evaluation of Copper Concentration

The application of electricity during the PBS buffer treatment with the copper device revealed a significant increase in copper. The copper concentration in PBS buffer increased at a greater rate through time than was the case with no treatment (*p* < 0.001). Furthermore, when the complete treatment or treatment with electricity was applied, the copper concentration was 10 times higher than it was in the treatment without electricity (*p* < 0.001) (Figure 1). Copper concentrations in treatments using electricity at different exposure times were 0 ppm (at 0 s), 195 ppm (at 5 min), 397 ppm (at 15 min), and 616 ppm (at 30 min). By contrast, treatments without the application of electricity resulted in a copper concentration of 0 ppm (0 s), 8 ppm (5 min), 16 ppm (15 min), and 24 ppm (30 min).

### 2.2. The Efficacy of Copper Ion Treatment: Estimation of Live Bacterial Load 

MAP at three concentrations (10^6^, 10^4^, and 10^2^ cells/mL) was able to grow after different treatments. However, there were differences in the number of cells recovered after treatment, depending on the time of treatment and the initial inoculum. Therefore, diluting inoculum led to a reduction in load of the bacterium in culture medium and an increase in time to detection (TTD) value, in which this change ranged between 8.72 and 3.68 (cells/mL), and between 0.4 and 16 days, respectively (Table 1).

Copper ion treatment resulted in a decreasing number of viable MAP detected in the BACTEC–MGIT 960 culture system, with a range of decay from 1.98 to 4.74 Log_10_ cells, immediately after treatment (moving from the most MAP-concentrated buffer treated to the most diluted buffer treated) (Table 1).

### 2.3. The Efficacy of Copper Ion Treatment: Estimation of Bacterial Load by the qPCR Approach

Bacterial load via qPCR estimation immediately after treatment showed a significant decrease after 5 min (*p* < 0.01), estimating a 2-log decrease in bacterial load in the MAP inoculated buffer, and reaching 4-log at 15 min (*p* < 0.001). However, in all samples that underwent the complete treatment, evidence of MAP DNA in the buffer disappeared at 30 min (*p* < 0.001) (Table 2). There was a statistically significant effect of the interaction between the treatment and time (*p* < 0.001) on bacterial load in PBS buffer. 

No significant difference in bacterial load was observed between copper treatment without electricity and no treatment (*p* = 0.491). However, there was a decrease in bacterial load from 10^6^ to 10^4^ MAP/mL observed in the copper treatment without electricity after 30 min of exposure (Table 2). Control treatments using steel did not show any differences between pre- and post-treatment in terms of bacterial load, except for the 30-min exposure, for which a decrease in MAP load was observed (data not shown), similar to that achieved by treatment with copper without electricity.

### 2.4. Correlation between Copper Concentration and Bacterial Load

There was a clear inverse relationship between the MAP load in the PBS buffer, estimated by IS*900* qPCR, and the copper concentration as the exposure time increased (*ρ* = −0.99) (Figure 2), while for the MAP load estimated in BACTEC–MGIT 960 culture, there was a more moderate, inverse relationship (*ρ* = −0.86) (Figure 2).

### 2.5. Physicochemical Properties of the Treated Buffer

We observed an increase in both pH and conductivity, and a decrease in oxygen concentration, after the application of copper ion treatment on MAP contaminated PBS buffer. The temperature remained constant (between 21 and 23 °C) in all treatments. When electricity was not included in the copper treatment, no significant differences were observed in the physicochemical properties of pre- and post-treated buffer (Table 3). In addition, an increase in pH, a constant conductivity rate, a decrease in oxygen consumption, and no change in temperature (data not shown) were also observed in the control treatments using steel. A similar situation was observed for the complete treatment of the uncontaminated and treated PBS buffer (data not shown).

## 3. Discussion

The reported efficacy of copper ions in the control of MAP amounts to an interesting scientific finding and suggests a potential tool for the control of this important potential zoonotic pathogen [21]. The idea is to recommend the application of this decontaminating principle to block an important infection transmission route from the most infectious animals (cows) to the most susceptible animals (calves) via milk consumption [24]. 

However, the fact that some MAP cells were able to tolerate this novel treatment [14,15] raises more questions than answers, which highlighted the need for us to continue exploring for more specifics regarding the efficacy of this innovative copper-based treatment. In fact, genotypic and phenotypic differences among MAP strains (S, C, and bison types) might result in resistance of the bacterium to the copper treatment. In addition, the reactivity of MAP to copper treatment is more unpredictable in naturally contaminated samples, as a recent study depicted that all three MAP strains of S, C, bison types existed in sheep milk samples taken from bulk tank milk and individual samples [25].

Primarily, we looked for more details regarding the efficacy of this novel treatment. We considered the copper concentration as a proxy for the release of copper ions, as has been described [26]. In addition, we tested this treatment for times greater than 30 min; however, at longer exposure times, the damage did not change, compared with that detected at 30 min. Exposure times greater than 30 min generated a significant increase in the temperature of the liquid matrix where the bacteria were suspended (PBS). This would be an undesirable effect, since the rise in temperature may confuse the deleterious effect of copper ions on the pathogen. Despite the robustness of the MAP detection and quantification system, we also need to consider some small limitations, such as the unknown origin of the detected DNA by qPCR either from a living or dead cell or whether it was free DNA. Additionally, the number of viable bacteria can be over- or underestimated in the culture system used, when the estimation between values of inoculum is lower than 10^2^ or higher than 10^5^.

In the present study, we could not totally reduce MAP load in a PBS buffer, since some MAP cells were able to survive and grew in culture media, even at the lowest MAP concentration and during the longest time of exposure to more than 600 ppm of copper concentration. Regarding copper concentrations, although we found a very marked negative correlation between the concentration of copper in PBS buffer and MAP load estimated directly by qPCR, this relationship was more moderate in culture. The interpretation that we can draw from this is that MAP must have strategies to resist the attack of copper, firstly due to the tendency of MAP cells to cluster, thereby protecting cells in the center [19], its lipid-rich, hydrophobic cell wall of MAP, and a nonreplicating, inactive, dormant, spore-like state [27]. Additionally, some mechanisms associated with the cellular repair or, even more likely, related to a topical intracellular tolerance mechanism associated with homeostasis genes activated by copper (chaperones, storage proteins, and efflux proteins), already described in other bacteria [28,29,30], subsequently allowed some MAP cells to survive and multiply in the culture medium. Furthermore, this copper-specific MAP tolerance may also be associated with its own nonspecific bacterial resistance characteristics [16,27,31].

Perhaps the most important finding that this study has revealed is a more detailed understanding of the efficacy of copper ion treatment on MAP. This includes the concentration of the metal that was reached in the liquid matrix where the bacteria were suspended, although the copper level reached was not sufficient to overcome the potential defensive mechanisms of MAP against the aggression of copper ions. The aforementioned situation is consistent with a study in which, for copper to be completely effective against *Staph. aureus* Methicillin resistance, levels as high as 1000 ppm were reported [32]. 

When the bacterial load was estimated by qPCR, our results were consistent with those of previous findings [14,15], but dissimilar results were observed when the estimation was performed in BACTEC MGIT culture. We must remember the principle of detection governing both diagnostic strategies. On the one hand, in a buffer, the mere presence of DNA is reported as a positive result. This can come from a viable cell, dead cell, or exist as free DNA [33,34,35]. The detection capacity is higher for culture than direct PCR, since, for MAP, culture provides an ideal growth condition in terms of nutrition and time, increasing the chances for its detection [34,36]. Therefore, MAP counts in culture were slightly higher than the qPCR determination. It must be highlighted that when the TTD indicator had the lowest quantity at different MAP inoculum diluted in PBS buffer, the amount of MAP in the BACTEC–MGIT culture system was estimated as very high, and this might be due to the limitations of the standard curve created for this purpose and the tendency of the bacterium to form clumps, which may affect the quantification accuracy of this counting method or MAP in culture.

The fact that, after treatment with copper ions, MAP DNA was not detected, strongly suggests that copper ions cause direct DNA damage in MAP cells, as has been reported elsewhere for other bacteria [7]. 

In relation to physicochemical properties, neither the increase in pH nor the decrease in oxygen concentration was associated with a decrease in MAP load in the buffer. Conversely, the stimulation of copper plates with an electric current caused a greater release of copper ions than that without an electric current. A recent study determined that electrically stimulated copper plates led to double the reduction in bacteria *(E. coli*), compared with electrically stimulated stainless-steel plates [22], which could be related to the production of free radicals such as hydroxyl ions and the production of reactive oxygen species (ROS) [37], favoring the formation of cuprous oxide and cupric oxide [38]. 

The increase in conductivity and the decrease in MAP load were also consistent with the high concentration of copper measured in the buffer after treatment. The strong negative correlation between the bacterial load of MAP and the concentration of copper confirms the synergistic effect between this metal and the electric current, which has a greater capacity to eliminate the bacteria, particularly at a longer exposure time, and which produces a greater release of copper ions, all of which may be due to the redox states [37,38]. 

Secondly, electric current may affect the orientation of membrane lipids and consequently the cell viability of bacteria [39]. A strong electric current can cause irreversible permeabilization of the cell membrane (depolarization of ion channels) and can even directly oxidize cellular components of bacteria [40,41]. Proof of this was the fact that the physicochemical properties of buffer underwent the same modifications after treatment with stainless steel under electrical current but without significant negative effects on MAP load. This lower antimicrobial activity of stainless steel in relation to copper has also been reported by other authors [3,7]. 

## 4. Materials and Methods

### 4.1. Preparation of Pure MAP Cultures 

The MAP ATCC 19698 strain was cultivated in a 7H9 liquid medium supplemented with 10% oleic acid–albumin–dextrose–catalase (OADC) (Becton Dickinson and Company, Sparks, MD, USA), 2 mg/L of mycobactin J (Allied Monitor, Fayette, MO, USA), and 5 mL/L of glycerol, for 1 month at 37 °C [36]. Once this pure culture reached an approximate concentration of 10^8^ cells/mL estimated by an optical density of 1 [36], serial 1:10 dilutions were made, to obtain the required concentration for the experimental challenge in phosphate-buffered saline (PBS).

### 4.2. Copper Ion Treatment Device

This consisted of a glass container (Pyrex® beaker, Lakewood, CO, USA) that contained 500 mL of buffer (PBS) inoculated with MAP, into which two high-purity copper plates (99%) were introduced. The copper plates were stimulated with a low voltage electrical current (24 V, 3 amps) to allow the active release of copper ions. A magnetic stirrer placed in the container enabled constant mixing during the treatment [14,15]. 

### 4.3. Estimation of Bacterial Load Using BACTEC–MGIT 960 Culture after the Application of Copper Ion Treatment

The concentration of MAP-spiked PBS buffer was adjusted to 10^6^, 10^4,^ and 10^2^ cells/mL dilutions. Then, the bacterial cells in the different dilutions were exposed to copper + electricity for the following periods of time: 0 min; 5 min; 15 min; 30 min. During the exposure time, the copper plates were stimulated as previously described to generate copper ions [14]. The copper concentration was estimated in the PBS buffer by atomic absorption spectrophotometry (AAS) at the end of each exposure time, and ppm was the measurement used. Each treatment was replicated three times. 

Subsequently, 100 µL aliquots of each treated MAP PBS buffer dilution were taken to be inoculated in the BACTEC–MGIT 960 liquid culture system to confirm whether MAP survival occurred, following the manufacturer’s instructions, with slight modifications. Briefly, the BACTEC–MGIT tubes were supplemented with 0.8 mL of ParaTB supplement (Becton Dickinson, Sparks, MD, USA) and 0.5 mL of egg yolk (Becton Dickinson, Sparks, MD). As it was not a clinical sample, a contaminating microbiota load was not expected. Therefore, antibiotics were not included, since these could affect the analytical sensitivity of culture methods [42]. Positive MGIT tubes were subjected to DNA extraction and purification, to then confirm MAP identity using a qPCR protocol [43]. 

The MAP load was estimated by the BACTEC–MGIT 960 culture system, which used the TTD value, an indicator that reports a significant growth of MAP to be categorized as a positive sample [36]. The TTD indicator of the BACTEC–MGIT culture system was converted to an inoculum number of MAP using a standard curve created by the TTD values for serial dilutions of MAP cell suspensions, according to the following equation: Log_10_ CFU = Span × ^e^(−K × TTD) + Plateau (where Span = 8.034; plateau = 0.9361 and K = −0.06852). The TTD values were converted to estimated numbers of MAP organisms by a standard curve [31,36]. The TTD values > 40 days were interpreted as ≤ 10 MAP organisms, and TTD values < 10 days were interpreted as ≥ 10^6^ MAP organisms.

### 4.4. Estimation of Bacterial Load Using qPCR after the Application of Copper Ion Treatment

For these experiments, MAP suspension at a concentration of 10^6^ cells per mL was inoculated into PBS buffer and later was subjected to a simultaneous copper and electricity treatment testing different exposure times of 0, 5, 15, and 30 min. 

As an electricity control, the copper treatment was applied without electricity, under the same exposure times as above. The experiments were carried out with three replicates for each treatment. 

Three negative controls were used including a MAP-inoculated PBS buffer that was not treated with copper ions; a MAP-inoculated PBS buffer treated with stainless steel and electricity (24 V) (metallic control); another PBS buffer that was not contaminated with MAP inoculum and treated with copper and electricity (24 V). The treatment exposure time for these was also 0, 5, 15, and 30 min., with three replicates also being carried out for each treatment.

MAP-inoculated PBS buffer samples before and after treatment were directly subjected to DNA extraction and purification, according to a published protocol [43]. The number of MAP cells from MAP-inoculated PBS buffer before and after copper ion treatment was estimated by the genomic equivalence principle, using a qPCR (Roche LightCycler 2.0 qPCR) to generate a standard curve, as previously described [44]. 

The genomic equivalence principle uses a standard curve based on the concentration of MAP DNA measured on a Nanoquant spectrophotometer (TECAN Group, Männedorf, Schweiz) adjusted to a 10^8^ dilution, the number of copies of the IS*900* target gene, and the reference of the molecular weight of the genome of MAP ATCC strain 19698. The copy numbers of the target region were expressed as MAP-specific bacterial cell equivalents (bce), according to a published equation [45].

The Roche 2.0 system for running qPCR to estimate MAP load and to confirm MAP identity was set according to a published protocol [43]. The qPCR reaction consisted of 5 µL DNA template, 10 µL of 2× TaqMan Universal Master Mix (Roche, Mannheim, Germany), 0.2 µM primers, and 0.1 µM probe, making up a final volume of 20 µL. The sequences of the primers used to amplify a 63-nucleotide fragment of the IS*900* gene were 5’-gacgcgatgatcgaggag-3’ (L) and 5’-gggcatgctcaggatgat-3’ (R). The amplification conditions were as follows: One cycle at 95 °C for 10 min; 45 cycles with three steps of 95 °C for 10 s, 60 °C for 30 s, and 72 °C for 1 sec; a final cooling step at 40 °C for 30 s. The negative (only PCR water) and positive (MAP ATCC 19698) controls were used for both the DNA extraction protocol and PCR reaction.

### 4.5. Evaluation of the Change in Physicochemical Composition after the Application of Copper Ion Treatment

To evaluate variations in the physicochemical properties of the treated and untreated MAP copper PBS buffer, the following analyses were carried out: (i) pH, employing the potentiometric method, which records hydrogen ion activity using a glass electrode (Orion, model 420A, Texas City, TX, USA); (ii) conductivity, through the electrometric method, which was used to estimate the total content of ionic constituents, using an electrode (Hanna Instrumental, edge ™, Woonsocket, RI, USA); (iii) oxygen consumption in the liquid matrix, employing an oximeter (Oxy 730 Inolab, Burladingen, Germany) to determine the dissolved oxygen in an aqueous solution; (iv) temperature, which was recorded using a thermometer to evaluate if there were any thermal changes in the buffer for each of the exposed treatments. Each determination was replicated 3 times.

### 4.6. Statistical Analysis 

A repeated-measures ANOVA test was used to determine significant differences between the effect of each response variable (copper concentrations, bacterial load using IS*900* qPCR, and physicochemical properties) before and after copper treatment, and with and without electricity treatments, using the factor time as the within-subject variable and the factor treatment as the intersubject variable. As this type of ANOVA does not allow us to check whether requirements (homogeneity and distribution) are met with the model residuals, we built a linear model and thus checked the homogeneity of variance using the Levene test and the distribution of our model using the Shapiro–Wilk test. Finally, to determine those treatments in which there were significant differences at each exposure time in comparison with no treatment, multiple post hoc comparisons with *t*-test, adjusted with Bonferroni test, were made [46]. 

Using a Pearson correlation, we estimated the relationship between MAP loads and copper concentration in PBS buffer after the complete copper ion treatment.

All statistical analyses and graphs were performed using R statistical software version 3.1.2. R 2015 (core development team). Differences with *p* < 0.05 were considered significant.

## 5. Conclusions

Under the experimental conditions in which this study was carried out, it can be concluded that, while the effectivity of copper ions was significant enough to reduce the number of MAP cells, it was not complete. It was confirmed that, even at low concentrations, the treated pathogen can grow in culture. Finally, the higher the conductivity, the lower the MAP load, which was also consistent with the high concentration of copper measured in the buffer after treatment. This shows that copper concentration is a key element in increasing the efficacy of this MAP control treatment to the expected level.

## Figures and Tables

**Figure 1 pathogens-11-00272-f001:**
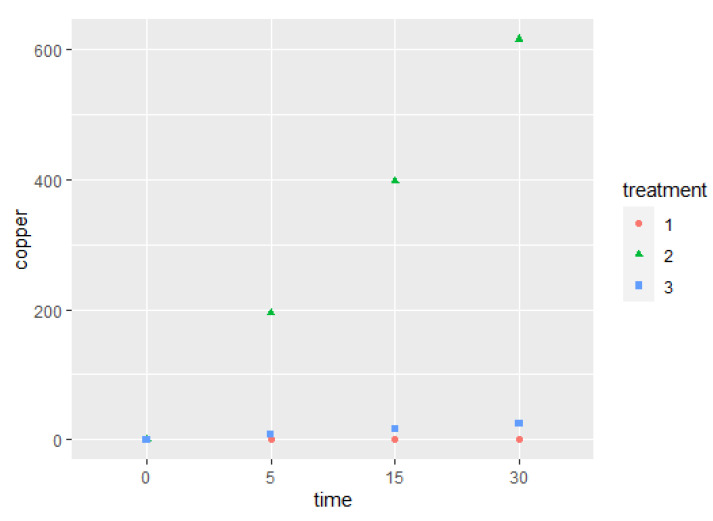
Copper concentration (ppm) determined by AAS in PBS buffer artificially contaminated with MAP without treatment (1) and after the application of two treatment strategies: copper ions with electricity (2) and only copper ions (3) for exposure times ranging between 0 and 30 min.

**Figure 2 pathogens-11-00272-f002:**
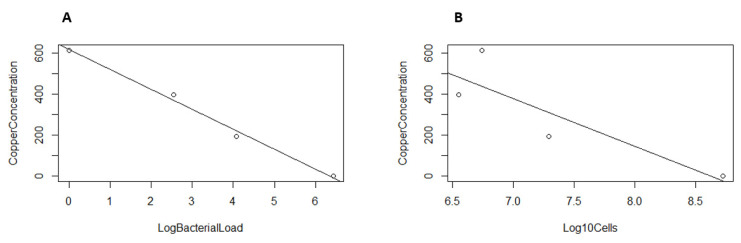
(**A**) Correlation between the estimation of the bacterial MAP concentration by qPCR (LOG (GE/UL)) and the copper concentration (PPM) released after the treatment with copper and electricity; (**B**) correlation between the estimation of the bacterial MAP concentration in liquid BACTEC–MGIT 960 culture (Log10 Cells) and the copper concentration (PPM) released after the treatment with copper and electricity.

**Table 1 pathogens-11-00272-t001:** MAP load estimation in the BACTEC–MGIT 960, according to TTD values and MAP dilution inoculum for each copper treatment exposure time.

Cu Exposure Time (min)	TTD Days (Mean ± SD)			Log_10_ Cells (Mean ± SD)		
	10^6^	10^4^	10^2^	10^6^	10^4^	10^2^
	MAP Dilution (Cells/mL) (Mean ± SD)			MAP Dilution (Cells/mL) (Mean ± SD)		
0	0.46 ± 0.00	0.67 ± 0.00	1.05 ± 0.01	8.72 ± 0.00	8.61 ± 0.00	8.42 ± 0.01
5	3.63 ± 3.35	8.34 ± 1.59	15.4 ± 0.02	7.29 ± 1.45	5.49 ± 0.50	3.74 ± 0.01
15	5.25 ± 1.06	7.61 ± 1.85	16.6 ± 4.16	6.55 ± 0.40	5.73 ± 0.60	3.57 ± 0.74
30	4.75 ± 0.35	8.80 ± 0.29	16.1 ± 4.74	6.74 ± 0.13	5.34 ± 0.09	3.68 ± 0.88

TTD: time to detection (days); Log_10_ cells MAP load in MGIT; SD: standard deviation

**Table 2 pathogens-11-00272-t002:** MAP load estimation using the qPCR approach, according to the genomic equivalence principle for each exposure time and type of treatment (no treatment, and copper with and without electricity).

	MAP	Load	
Cu Exposure Time (min)	No TT	Complete Cu TT	Cu TT w/o E
	(mean ± SD)	(mean ± SD)	(mean ± SD)
0	3.34 × 10^6^ ± 340,293 ^a^	2.71 × 10^6^ ± 74,833 ^a^	4.97 × 10^6^ ± 79,302 ^a^
5	3.46 × 10^6^ ± 43,204 ^a^	1.20 × 10^4^ ± 499 ^b^	5.18 × 10^5^ ± 10,801 ^a^
15	3.87 × 10^6^ ± 101,980 ^a^	3.46 × 10^2^ ± 4.92 ^c^	3.43 × 10^5^ ± 18,018 ^a^
30	3.8 × 10^6^ ± 82,596 ^a^	0.00 × 10^0^ ± 0.00 ^c^	7.82 × 10^4^ ± 589 ^a^

Superscript indicates statistically significant difference between each treatment: (a) no significant differences; (b) significant differences *p* < 0.05; (c) significant differences *p* < 0.005. MAP load: Expressed in bce mL^−1^. No TT: no treatment with copper ions in the MAP-contaminated buffer sample. Complete Cu TT: copper plates immersed in the buffer sample and stimulated with a low voltage (24 V) electrical current (3 amperes). Cu TT w/o E: copper plates immersed in the buffer sample without the application of the electrical current.

**Table 3 pathogens-11-00272-t003:** Average values for pH, electrical conductivity (EC), and estimated oxygen concentration ([O_2_]) in buffer artificially contaminated with MAP before and after three treatment strategies (no treatment, and copper with and without electricity) for a given period of time.

Cu Exposure Time (min)	pH with No TT	pH with Complete Cu TT	pH with Cu TT w/o E	EC with No TT	EC with Complete Cu TT	EC with Cu TT w/o E	[O_2_] with No TT	[O_2_] with Complete Cu TT	[O_2_] with Cu TT w/o E
	(mean ± SD)	(mean ± SD)	(mean ± SD)	(mean ± SD)	(mean ± SD)	(mean ± SD)	(mean ± SD)	(mean ± SD)	(mean ± SD)
0	7.5 ± 0.04	7.5 ± 0.04	7.5 ± 0.07	4.7 ± 0.09	4.7 ± 0.08	4.8 ± 0.02	7.3 ± 0.12	7.4 ± 0.39	7.6 ± 0.02
5	7.5 ± 0.02	11.0 ± 0,09	7.4 ± 0.02	4.7 ± 0.06	4.8 ± 0.07	4.9 ± 0.03	7.8 ± 0.18	4.3 ± 1.38	7.5 ± 0.03
15	7.6 ± 0.04	11.8 ± 0.09	7.2 ± 0.02	4.7 ± 0.07	5.5 ± 0.08	4.9 ± 0.03	7.6 ± 0.21	3.3 ± 1.50	7.3 ± 0.03
30	7.6 ± 0.04	11.9 ± 0.04	7.2 ± 0.03	4.7 ± 0.04	5.8 ± 0.28	4.9 ± 0.02	7.5 ± 0.21	2.5 ± 1.31	7.0 ± 0.04

EC: electrical conductivity expressed in mS cm^−1^. [O_2_]: oxygen concentration expressed in mg l^−1^. no TT: no treatment with copper ions in the MAP-contaminated buffer sample. Complete Cu TT: copper plates immersed in the buffer sample and stimulated with a low voltage (24 V) electrical current (3 amperes). Cu TT w/o E: copper plates immersed in the buffer sample without the application of the electrical current.

## Data Availability

The datasets used and/or analyzed during the current study are available from the corresponding author on reasonable request.

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
