# Peer review of "More Insights about the Efficacy of Copper Ion Treatment on Mycobacterium avium subsp. paratuberculosis (MAP): A Clue for the Observed Tolerance"

_pathogens, 2022, doi:10.3390/pathogens11020272_

Round 1

Reviewer 1 Report

The work is focused on basic knowledge about the influence of copper on MAP in vitro. The authors have already attempted to use copper to devitalize MAP in vivo. Three papers are cited in the manuscript (2018, 2020, 2021), one more (2021) is not cited. In these articles, the authors did not demonstrate complete protection of calves from MAP infection by treating milk with copper.

The results of the experiments describe the release of Cu ions from copper plates by an in vitro electric current in a buffer. With duration of action, MAP viability decreases, but complete MAP devitalization has not been achieved.

The authors must introduce only relevant citations in the introduction and define their intention as the design of the proposed experiments with a clear hypothesis of what new knowledge they want to verify in the experiments. To verify the hypothesis, the methodology and description of the achieved results must be focused. The results must be confronted in the discussion only with the data given in the introduction. It must be clear from the discussion whether the hypothesis can be considered verified or whether it has not been proven to be correct. Nothing else belongs in the discussion. The results should indicate their importance for basic knowledge and for practice. The conclusions of this work present speculations based on the results of other works published by the authors. However, this speculation is not related to this article. The work must be processed simply, using all known information on the topic, but without unnecessary ballast.

Author Response

Answer: we appreciate the reviewer’s comment. After the corrections made from comments and suggestion by all the reviewers, we believe now, the corrected version of our manuscript achieve the standard suggested by reviewer 1.

We have carefully considered the comments of all reviewers on the Manuscript ID: pathogens-1581796 and now return to you an improved manuscript for publication consideration. I hope that the content of this revised version fulfils the scope requirements of your prestigious journal.

Reviewer 1
The work is focused on basic knowledge about the influence of copper on MAP in vitro. The authors have already attempted to use copper to devitalize MAP in vivo. Three papers are cited in the manuscript (2018, 2020, 2021), one more (2021) is not cited. In these articles, the authors did not demonstrate complete protection of calves from MAP infection by treating milk with copper.
The results of the experiments describe the release of Cu ions from copper plates by an in vitro electric current in a buffer. With duration of action, MAP viability decreases, but complete MAP devitalization has not been achieved.
The authors must introduce only relevant citations in the introduction and define their intention as the design of the proposed experiments with a clear hypothesis of what new knowledge they want to verify in the experiments. To verify the hypothesis, the methodology and description of the achieved results must be focused. The results must be confronted in the discussion only with the data given in the introduction. It must be clear from the discussion whether the hypothesis can be considered verified or whether it has not been proven to be correct. Nothing else belongs in the discussion. The results should indicate their importance for basic knowledge and for practice. The conclusions of this work present speculations based on the results of other works published by the authors. However, this speculation is not related to this article. The work must be processed simply, using all known information on the topic, but without unnecessary ballast.
Answer: we appreciate the reviewer’s comment. After the corrections made from comments and suggestion by all the reviewers, we believe now, the corrected version of our manuscript achieve the standard suggested by reviewer 1.

Reviewer 2 Report

Overview: The manuscript entitled "More Insights about the Efficacy of Copper Ion Treatment on Mycobacterium avium subsp. paratuberculosis (MAP). A Clue for the Observed Tolerance" sounds like an interesting study about the inhibitory impacts of copper on MAP viability. This inhibitory effect could be exploited in treatment of infections caused by MAP. However, more studies are needed to be done in order to verify the efficacy of the assay in naturally MAP-contaminated specimens considering the matrix of sample and differences existed among the types of MAP strain (S, C, bison).

To improve the quality of the present manuscript, some modifications could be implemented according to the following concerns:

1) Manuscript needs to be polished grammatically. There are some long sentences that are confusing.

2) Please summarize the abstract to maximum 200 words.

3) Line 20: Please name the buffer!

4) Line 20: Change "some survival was also observed" to "although, the sustainability of MAP against copper toxicity was also reported in several studies ".

5) Line 21: This sentence doesn't sound well here (it is suspended as ended to full stop. It seems that you connected it to methodology! right?) "With the aim of deepening our understanding of the efficacy of this novel treatment". I suggest to merge this sentence to the next one as follows: Accordingly, this study aimed to evaluate the inhibitory impacts of copper on MAP cells suspended in PBS considering the physico-chemical properties of this buffer on susceptibility or resistance of MAP cells to copper.

6) Line 22: In methodology section, please briefly explain which essential methods enabled you to measure the efficiency of copper against MAP.

7) Line 26: It should be "grow"

8) Line 26-28: Merge the two sentences as follows: However, longer exposure of MAP to copper could not only increase the concentration of copper ion in the buffer, but also decrease the load of bacterium measured by qPCR analysis in samples significantly, compared to that of MGIT culture.

9) Line 38: Change to this: Copper is a chemical element that, at low concentration, plays an essential role in animal and plant cell metabolism.

10) Line 57: one of the most fastidious

11) Line 58: resistant to what? did you mean drug/antibiotic-resistant?

12) Line 59: MAP is a zoonotic pathogen. I suggest you to mention it in this sentence.

13) Line 59: Transfer "in a liquid matrix" to  after "copper ions" as follows: In a more novel way, it has been shown that, copper ions, in a liquid matrix,...

14) Line 75: resistant to what? Please change "one of the most animal resistant pathogens" to " one of the most drug-resistant zoonotic pathogens, MAP.

15) Line 81: What is the complete treatment? Did you mean treatment with electricity?

16) Figure 1: Please write the unites of concentration (ppm) and time (min) in parentheses in horizontal and vertical axes labels.

17) Line 95: Please define TTD here, as this is the first time that you explain it.

18) Line 95: remove article "the" from behind of "inoculum" and "TTD".

19) Line 95: As I understood, you mentioned that by concentrating inoculum, lower TTD would be expected! But the second part seems ambiguous "the higher the estimated bacterial load in the culture medium, where the TTD value varied from 0.4 to 16 days"! Did you mean, "therefore diluting inoculum led to a reduction in the load of bacterium in culture medium and an increase in TTD value, in which this changes ranged between 8.72-3.68 (cells/mL) and 0.4-16 days respectively"?   

20) Line 108: Does "complete treatment" mean treatment with copper and electricity? You should explain it somewhere in the manuscript.

21) Line 121: Could you define the superscript a, b, and c?

22) Figure 2: please write the unit of concentration of copper at vertical axes labels.

23) Line 142: Change "When copper treatment did not include the application of electricity" to "When electricity was not included in the copper treatment"

24) Line 161: change "animal pathogen" to "zoonotic pathogen" and cite it with the following reference: A. SL, Mura M, Tanda F, Lissia A, Solinas A, Fadda G, Zanetti S. 2001. Identification of Mycobacterium avium subsp. paratuberculosis in Biopsy Specimens from Patients with Crohn’s Disease Identified by In Situ Hybridization. J Clin Microbiol 39:4514–4517.

25) Line 168: I suggest to add the following hypothesis after your statement: "In fact, genotypic and phenotypic differences among MAP strains (S, C, and bison types) might result in resistance of the bacterium to the copper treatment.  In addition, the reactivity of MAP to copper treatment is more unpredictable in naturally contaminated samples, as a recent study depicted that all three MAP strains of S, C, bison types existed in sheep milk samples taken from bulk tank milk and individual samples". Citation: (Hosseiniporgham,S.; Biet, F.; Ganneau, C.; Bannantine, J.P.; Bay, S.; Sechi, L.A. A Comparative Study on the Efficiency of Two Mycobacterium avium subsp. paratuberculosis (MAP)-Derived Lipopeptides of L3P and L5P as Capture Antigens in an In-House Milk ELISA Test. Vaccines 2021, 9, 99.)

26) Line 202: aforementioned is an adjective, you should use noun after it. Please rewrite line 202-204.

27) Line 207-208: Please rewrite this sentence "We must remember the nature of detection governing both diagnostic strategies". what is the nature of detection?

28) Line 210-213: please rewrite this sentence.

29) Line 215: The order is confusing: Change it to this: It must be highlighted that when the TTD indicator had the lowest quantity at different MAP inoculum diluted in PBS buffer, the amount of MAP in BACTEC-MGIT culture system was estimated very high and this might be due to the limitations of the standard curve created for this purpose and the tendency of the bacterium to form clumps...."

30) Line 288: Correct this sentence as follows: "MAP suspension at concentration of 106 cells per mL was inoculated into PBS buffer and later was subjected to a simultaneous copper and electricity treatment testing different exposure times of 0, 5, 15, and 30 minutes. 

31) Line 292: remove done (use one of them (done or carried out))

32) Line 294: Correct this: "Three negative controls were used including:"

33) Line 295: "was not challenged with"? do you mean "was not treated with"?

34) Line 296-297: Change it to "PBS buffer that was not contaminated with MAP inoculum and treated with...."

    1.  

  1.  

Author Response

Reviewer 2

Overview: The manuscript entitled "More Insights about the Efficacy of Copper Ion Treatment on Mycobacterium avium subsp. paratuberculosis (MAP). A Clue for the Observed Tolerance" sounds like an interesting study about the inhibitory impacts of copper on MAP viability. This inhibitory effect could be exploited in treatment of infections caused by MAP. However, more studies are needed to be done in order to verify the efficacy of the assay in naturally MAP-contaminated specimens considering the matrix of sample and differences existed among the types of MAP strain (S, C, bison).

To improve the quality of the present manuscript, some modifications could be implemented according to the following concerns:

1) Manuscript needs to be polished grammatically. There are some long sentences that are confusing.

2) Please summarize the abstract to maximum 200 words.

Answer: the abstract has been summarized to 200 words, as suggested. See page (P) 1 line (L)16 of new version (NV).

3) Line 20: Please name the buffer!

Answer: in order to summarize the abstract, it is better to name the PBS in the material and methods part. P7, L 257 NV.

4) Line 20: Change "some survival was also observed" to "although, the sustainability of MAP against copper toxicity was also reported in several studies ".

Answer: changed as suggested. P 1, L19-20 NV.

5) Line 21: This sentence doesn't sound well here (it is suspended as ended to full stop. It seems that you connected it to methodology! right?) "With the aim of deepening our understanding of the efficacy of this novel treatment". I suggest to merge this sentence to the next one as follows: Accordingly, this study aimed to evaluate the inhibitory impacts of copper on MAP cells suspended in PBS considering the physico-chemical properties of this buffer on susceptibility or resistance of MAP cells to copper.

Answer: changed as suggested. P 1. L20-22 NV.

6) Line 22: In methodology section, please briefly explain which essential methods enabled you to measure the efficiency of copper against MAP.

Answer: we regret to inform you that there is no more room to extra words to fulfil the 200 words

7) Line 26: It should be "grow"

Answer: the word “grew” was deleted to reach the number of the words needed

8) Line 26-28: Merge the two sentences as follows: However, longer exposure of MAP to copper could not only increase the concentration of copper ion in the buffer, but also decrease the load of bacterium measured by qPCR analysis in samples significantly, compared to that of MGIT culture.

Answer: the sentence was merged and summarized, as suggested. P 1, L 25-31 NV

9) Line 38: Change to this: Copper is a chemical element that, at low concentration, plays an essential role in animal and plant cell metabolism.

Answer: changed as suggested. P 1, L35-36 NV.

10) Line 57: one of the most fastidious

Answer: changed as suggested. P2, L55-56 NV

11) Line 58: resistant to what? did you mean drug/antibiotic-resistant?

Answer: the sentence has been rewriting for clarity. P2, L56-57 NV.

12) Line 59: MAP is a zoonotic pathogen. I suggest you to mention it in this sentence.

Answer: the sentence has been included, as suggested. P2, L58 NV.

13) Line 59: Transfer "in a liquid matrix" to  after "copper ions" as follows: In a more novel way, it has been shown that, copper ions, in a liquid matrix,...

Answer: the sentence was transferred, as suggested. P2, L54 NV.

14) Line 75: resistant to what? Please change "one of the most animal resistant pathogens" to " one of the most drug-resistant zoonotic pathogens, MAP.

Answer: changed as suggested. P2, L73 NV

15) Line 81: What is the complete treatment? Did you mean treatment with electricity?

Answer: Yes, treatment with electricity. The word “electricity” has been included in the sentence for clarity. P2, L79 NV.

16) Figure 1: Please write the unites of concentration (ppm) and time (min) in parentheses in horizontal and vertical axes labels.

Answer: corrected as suggested. See Fig 1 NV

17) Line 95: Please define TTD here, as this is the first time that you explain it.

Answer: TTD was defined as suggested. P3, L96 NV.

18) Line 95: remove article "the" from behind of "inoculum" and "TTD".

Answer: The article “the” was removed, as suggested. P3, L95-96 NV.

19) Line 95: As I understood, you mentioned that by concentrating inoculum, lower TTD would be expected! But the second part seems ambiguous "the higher the estimated bacterial load in the culture medium, where the TTD value varied from 0.4 to 16 days"! Did you mean, "therefore diluting inoculum led to a reduction in the load of bacterium in culture medium and an increase in TTD value, in which this changes ranged between 8.72-3.68 (cells/mL) and 0.4-16 days respectively"?   

Answer: changed as suggested for clarity. P3, L94-97 NV

20) Line 108: Does "complete treatment" mean treatment with copper and electricity? You should explain it somewhere in the manuscript.

Answer: Yes, treatment with electricity. This has been already explained in P2, L79 NV.

21) Line 121: Could you define the superscript a, b, and c?

Answer: corrected as suggested. See Table 2 NV

22) Figure 2: please write the unit of concentration of copper at vertical axes labels.

Answer: corrected as suggested. See Fig 2 NV

23) Line 142: Change "When copper treatment did not include the application of electricity" to "When electricity was not included in the copper treatment"

Answer: changed as suggested. P5, L144-145 NV.

24) Line 161: change "animal pathogen" to "zoonotic pathogen" and cite it with the following reference: A. SL, Mura M, Tanda F, Lissia A, Solinas A, Fadda G, Zanetti S. 2001. Identification of Mycobacterium avium subsp. paratuberculosis in Biopsy Specimens from Patients with Crohn’s Disease Identified by In Situ Hybridization. J Clin Microbiol 39:4514–4517.

Answer: changed as suggested. P6, L163-164 NV.

25) Line 168: I suggest to add the following hypothesis after your statement: "In fact, genotypic and phenotypic differences among MAP strains (S, C, and bison types) might result in resistance of the bacterium to the copper treatment.  In addition, the reactivity of MAP to copper treatment is more unpredictable in naturally contaminated samples, as a recent study depicted that all three MAP strains of S, C, bison types existed in sheep milk samples taken from bulk tank milk and individual samples". Citation: (Hosseiniporgham,S.; Biet, F.; Ganneau, C.; Bannantine, J.P.; Bay, S.; Sechi, L.A. A Comparative Study on the Efficiency of Two Mycobacterium avium subsp. paratuberculosis (MAP)-Derived Lipopeptides of L3P and L5P as Capture Antigens in an In-House Milk ELISA Test. Vaccines 2021, 9, 99.)

Answer: the suggested paragraph has been included, as suggested. P6, L170-174 NV.

26) Line 202: aforementioned is an adjective, you should use noun after it. Please rewrite line 202-204.

Answer: the reviewer is correct. Hence, the word “situation” has been included, as suggested. P6, L 208 NV.

27) Line 207-208: Please rewrite this sentence "We must remember the nature of detection governing both diagnostic strategies". what is the nature of detection?

Answer: the reviewer is correct. The word “nature” has been changed by the word “principle”, as suggested. P6, L213 NV.

28) Line 210-213: please rewrite this sentence.

Answer: the sentence has been rewritten, as suggested. P7, L216-218 NV.

29) Line 215: The order is confusing: Change it to this: It must be highlighted that when the TTD indicator had the lowest quantity at different MAP inoculum diluted in PBS buffer, the amount of MAP in BACTEC-MGIT culture system was estimated very high and this might be due to the limitations of the standard curve created for this purpose and the tendency of the bacterium to form clumps...."

Answer: the sentence has been changed, as suggested. P7, L219-223 NV.

30) Line 288: Correct this sentence as follows: "MAP suspension at concentration of 106 cells per mL was inoculated into PBS buffer and later was subjected to a simultaneous copper and electricity treatment testing different exposure times of 0, 5, 15, and 30 minutes. 

Answer: the sentence has been changed, as suggested. P8, L292-294 NV.

31) Line 292: remove done (use one of them (done or carried out))

Answer: the word “done” was deleted, as suggested. P8, L296 NV.

32) Line 294: Correct this: "Three negative controls were used including:"

Answer: the sentence was corrected, as suggested. P8, L298 NV.

33) Line 295: "was not challenged with"? do you mean "was not treated with"?

Answer: the sentence was corrected, as suggested. P8, L298-299 NV.

34) Line 296-297: Change it to "PBS buffer that was not contaminated with MAP inoculum and treated with...."

Answer: the sentence was changed, as suggested. P8, L300-301 NV.

Reviewer 3 Report

The manuscript would be of value to the readership of Pathogens as it delivers the basis for alternative treatments to reduce MAP loads in colostrum and milk used for calves feeding, measuring the susceptibility of the pathogen to copper ions. The manuscript is well written and to improve its overall quality the authors are advised to make these minor revisions:

- Ln. 59-63: technical information like voltage and time exposure are adequately presented in the M&M paragraph and should be avoided in the introduction where the approach can be described in a more general way.  E.g. “Our experiment used a glass receptacle copper device in which copper plates of high purity (99 %) were immersed in a liquid matrix and were stimulated with electricity.”
- Ln. 95: the first appearance of the acronym TTD should be anticipated by the description: Time To Detection.
- Ln. 264-265: This information is already reported at ln. 257-258 hence this sentence could be cancelled.

Author Response

Reviewer 3

The manuscript would be of value to the readership of Pathogens as it delivers the basis for alternative treatments to reduce MAP loads in colostrum and milk used for calves feeding, measuring the susceptibility of the pathogen to copper ions. The manuscript is well written and to improve its overall quality the authors are advised to make these minor revisions:

- Ln. 59-63: technical information like voltage and time exposure are adequately presented in the M&M paragraph and should be avoided in the introduction where the approach can be described in a more general way.  E.g. “Our experiment used a glass receptacle copper device in which copper plates of high purity (99 %) were immersed in a liquid matrix and were stimulated with electricity.”

Answer: the reviewer is correct. The sentence has been simplified, as suggested. P2, L58-60 NV.

- Ln. 95: the first appearance of the acronym TTD should be anticipated by the description: Time To Detection.

Answer: TTD was defined as suggested. P3, L96 NV.

- Ln. 264-265: This information is already reported at ln. 257-258 hence this sentence could be cancelled.

Answer: the reviewer is correct. The sentence has been simplified, as suggested. P7, L269 NV.

Round 2

Reviewer 1 Report

Reviewer's comments were respected in part.

The manuscript does not have a sound working hypothesis. It is not explicitly stated that milk treatment could be practically useful for protecting calves from MAP.

There was information left in the discussion that was not directly related to the performed experiments.

Author Response

The manuscript does not have a sound working hypothesis. It is not explicitly stated that milk treatment could be practically useful for protecting calves from MAP.

Answers: the reviewer is correct. Now, we explicit a hypothesis in the introduction part. See page 2 (P), lines 67-71 (L) of new version (NV).

There was information left in the discussion that was not directly related to the performed experiments.

Answers: Most of the information contained in the discussion refers to experimental findings generated in this study. The information not directly related to the performed experiments were suggested to be included by the other reviewers

Reviewer 2 Report

The manuscript underwent considerable improvements. 

Good luck

Author Response

Answers: We thank the reviewer for her/his valuable inputs.

Reviewer 3 Report

I have only one concern regarding the "zoonotic" behavoir of MAP. At now this pathogen has been detected in different human matrices, in association with different chronic inflammatory diseases (IBDs, multiple sclerosis, ...) but there is no consensus in confirming MAP as the causative agent.

I'd suggest to use, as correctly done in line 58, the terminology "zoonotic potential" and amend the text accordingly at lines 73 and 163.

Kind Regards

Author Response

I have only one concern regarding the "zoonotic" behavoir of MAP. At now this pathogen has been detected in different human matrices, in association with different chronic inflammatory diseases (IBDs, multiple sclerosis, ...) but there is no consensus in confirming MAP as the causative agent.

I'd suggest to use, as correctly done in line 58, the terminology "zoonotic potential" and amend the text accordingly at lines 73 and 163.

Answers: corrected as suggested. See P2, L76 NV and P6, L 166-167 NV.